# Orientation-anchored Hyper-Gaussian for 4D Reconstruction from Casual Videos

**Junyi Wu**[1]     **Jiachen Tao**[1]     **Haoxuan Wang**[1]
**Gaowen Liu**[2]   **Ramana Rao Kompella**[2]   **Yan Yan**[1]*
[1]University of Illinois Chicago     [2]Cisco Research
https://github.com/adreamwu/OriGS

## Abstract

We present **Orientation-anchored Gaussian Splatting** (**OriGS**), a novel framework for high-quality 4D reconstruction from casually captured monocular videos. While recent advances extend 3D Gaussian Splatting to dynamic scenes via various motion anchors, such as graph nodes or spline control points, they often rely on low-rank assumptions and fall short in modeling complex, region-specific deformations inherent to unconstrained dynamics. OriGS addresses this by introducing a hyperdimensional representation grounded in scene orientation. We first estimate a **Global Orientation Field** that propagates principal forward directions across space and time, serving as stable structural guidance for dynamic modeling. Built upon this, we propose **Orientation-aware Hyper-Gaussian**, a unified formulation that embeds time, space, geometry, and orientation into a coherent probabilistic state. This enables inferring region-specific deformation through principled conditioned slicing, adaptively capturing diverse local dynamics in alignment with global motion intent. Experiments demonstrate the superior reconstruction fidelity of OriGS over mainstream methods in challenging real-world dynamic scenes.

## 1   Introduction

Videos offer a window into the continuous flow of forms, light, and transformations that weave our experience of space and time. Among various sources, *casually captured monocular videos* stand as the most ubiquitous and accessible. These videos are typically recorded by smartphones, handheld cameras, or consumer drones, spanning a broad spectrum of everyday scenes with diverse object compositions and unconstrained motion patterns. Recovering the underlying dynamic reality from such inputs remains a foundational pursuit in computer graphics and 3D vision [33], boosting applications in autonomous driving [72, 21], robotics [30, 53], and virtual/augmented reality [81, 10].

At its core, 4D reconstruction aims to recover the time-varying geometry, appearance, and motion of real-world environments from videos [82], bridging visual observations with underlying scene dynamics. 3D Gaussian Splatting (3DGS) [37] has recently emerged as a powerful solution for static scene reconstruction, offering a compact and expressive point-based representation that enables high-fidelity modeling and real-time rendering. Building on its success, substantial efforts have been devoted to extending 3DGS to dynamic scenes [49, 76, 71, 64]. These methods primarily capture temporal evolution by applying per-Gaussian deformations, such as translation and rotation, across time. Recent works are further pursuing robust reconstruction from casually captured videos by modeling deformation through various motion anchors. Representative approaches include learning motion bases and weighting coefficients [67], constructing discrete 3D node graphs [48, 39, 44], and employing continuous cubic Hermite splines with sparse control points [77, 54].

Yet, reconstructing scene dynamics from casually captured monocular videos remains challenging due to the complexity of local motion. Regions within the scene often exhibit diverse motion

---

*Corresponding Author.

39th Conference on Neural Information Processing Systems (NeurIPS 2025).

patterns, influenced by object articulation, interaction context, or progression through different action stages [61]. Although recent approaches incorporate explicit, structured deformation fields, they largely reduce complex dynamics to low-rank motion anchors, assuming that neighboring regions exhibit similar motion. While effective for smooth or rigid transformations, such deformation anchors are inherently fragile under unconstrained dynamics in real-world scenarios, where motion patterns may vary significantly across regions. Consequently, these formulations often struggle to capture complex local dynamics, resulting in spatial drift, structural fragmentation, or temporal inconsistency.

To address these challenges, we are motivated by the perspective that the diversity of local motion often reflects more than just positional or geometric variation; it suggests an underlying regularity in how different regions evolve over time. Intuitively, local deformations are not solely determined by spatial position or object shape, but also by how each part of a scene is *expected* to move within a broader dynamic context. Fortunately, while casual monocular videos lack explicit multi-view supervision, they do preserve object-centric motion cues that implicitly reflect these region-specific motion tendencies. Among them, we focus our exploration on a natural and grounded signal: *orientation*. In the physical world, while absolute position is sensitive to noise and scale ambiguity, orientation evolves under the smoother governance of local angular momentum and inertia [16, 27]. This makes it a more stable proxy for how motion is expected to unfold in complex dynamic scenes, indicating the forward tendencies across regions. In this work, we embrace orientation as a dynamic anchor for 4D reconstruction. Our key insight is to treat orientation as a structural signal that organizes how different parts of the scene deform and interact, thereby capturing coherent evolution.

Building on this perspective, we introduce **Orientation-anchored Gaussian Splatting** (**OriGS**), a novel framework for high-quality 4D reconstruction from casual monocular videos. Our OriGS comprises two synergistic components for dynamic modeling. ❶ We first construct a **Global Orientation Field** that captures long-range scene evolution by estimating and propagating principal forward directions over time via localized structure alignment. This orientation field provides stable oriented anchors that extend across space and time, which help organize diverse motion patterns in different regions. ❷ On top of this structured foundation, we propose **Orientation-aware Hyper-Gaussian**, a hyperdimensional representation to model complex local dynamics throughout the scene. We associate each Gaussian primitive with a unified probabilistic state defined over a structured manifold of time, space, geometry, and orientation. By conditioning this representation on the scene's evolving orientation field, we infer region-specific deformations through a principled conditioned slicing strategy. This allows OriGS to adaptively model how different parts of the scene deform over time and how their motions evolve in alignment with the underlying dynamic intent. Together, these two components enable OriGS to robustly reconstruct dynamic scenes from casual videos, bridging global coherence and local motion diversity in a unified and principled manner.

We validate the effectiveness of OriGS on a range of casual monocular videos, including DAVIS [59], OpenAI SORA [6], YouTube-VOS [74], and the DyCheck benchmark [19]. OriGS consistently recovers sharper geometry and more coherent motion compared to recent state-of-the-art methods, demonstrating its superior reconstruction fidelity in real-world dynamic scenes.

## 2 Related Works

**Dynamic 3D Representations.** Dynamic scene modeling has been widely studied by extending static 3D representations with deformation mechanisms [20, 9]. Neural Radiance Fields (NeRF) [51] represent scenes as volumetric functions, and are adapted to dynamic settings by learning deformation fields that warp canonical points over time [55, 60, 56, 17, 34, 70]. Later works introduce temporal embeddings [57, 40], or hybrid representations for faster rendering [2, 18, 8, 62]. However, their implicit volumetric nature limits real-time inference and structure-aware modeling. Recently, 3D Gaussian Splatting (3DGS) [37] has emerged as a powerful alternative for scene representation [4, 82]. Dynamic extensions incorporate deformation fields [49, 71, 76, 15, 3, 31], apply trajectory interpolation [41, 38, 77], or directly embed time into Gaussian primitives [14, 75]. Yet, lacking explicit guidance from underlying structures, these methods are generally incapable of faithfully recovering complex motion in real-world dynamic scenes.

**Hyperdimensional Gaussians.** Recent work has enhanced the expressiveness of Gaussian representations by embedding auxiliary signals into higher-dimensional spaces. 4DGS [75, 14] incorporates time to model spatio-temporal volumes, while N-DG [11] generalizes to a latent space embedding po-

sition, view direction, and material cues. To capture view-dependent effects, 6DGS [22, 24] integrates angular information and modulates opacity and color accordingly, and 7DGS [23] further extends this with temporal modeling. In this work, we explore hyperdimensional Gaussians as structured probabilistic states, modeling space, time, geometry, and orientation in a unified representation.

**4D Reconstruction from Casual Monocular Videos.** Recovering dynamic representations from casually captured monocular videos is highly ill-posed. Early efforts enhance NeRFs by specialized deformation fields [60, 55, 56], long-range feature aggregation [43], or point trajectory prediction [66]. While accounting for scene changes, they predominantly consider controlled settings with teleporting cameras or quasi-static scenes [19]. Recent advances shift toward Gaussian-based representations, leveraging 2D priors to enable reconstruction in the wild. Methods like Shape-of-Motion [67], Gaussian Marbles [64], MoSca [39], and MoDGS [46] guide per-Gaussian deformation through various motion anchors. Others adopt compact motion models, such as spline interpolation [54] or hierarchical structures [44], to trace temporal trajectories. Nonetheless, their reliance on low-rank motion priors limits the capacity to capture the complex, spatially varying dynamics present in real-world environments. In contrast, we propose to organize deformation within a global orientation field and condition dynamic modeling on a unified hyperdimensional representation, enabling coherent reconstruction under diverse motion patterns.

## 3 Preliminaries

**3D Gaussian Splatting.** 3DGS [37] represents scenes as a set of explicit 3D Gaussian primitives, each specified by a spatial center $\boldsymbol{\mu_p} \in \mathbb{R}^3$ and an anisotropic covariance matrix $\boldsymbol{\Sigma} \in \mathbb{R}^{3 \times 3}$:

$$G(\mathbf{x}) = \exp\left(-\frac{1}{2}(\mathbf{x} - \boldsymbol{\mu_p})^\top \boldsymbol{\Sigma}^{-1}(\mathbf{x} - \boldsymbol{\mu_p})\right), \tag{1}$$

where $\mathbf{x} \in \mathbb{R}^3$ denotes a 3D spatial location. To ensure positive semi-definiteness, $\boldsymbol{\Sigma}$ is factorized as $\boldsymbol{\Sigma} = \mathbf{RSS}^\top\mathbf{R}^\top$ in practice, where $\mathbf{S}$ is a diagonal scaling matrix and $\mathbf{R}$ is a rotation matrix aligning the Gaussian with the global coordinate system. Moreover, each 3D Gaussian associates an opacity value $\sigma \in \mathbb{R}$ and a color vector $\mathbf{c} \in \mathbb{R}^3$. During rendering, contributions from all Gaussians overlapping a pixel are composited via alpha blending to provide the final color.

In this work, we adopt 3DGS as a differentiable and compact representation for dynamic scenes, which can be efficiently rendered into images with a rasterization pipeline.

**3D Trajectories from Monocular Videos.** Given a monocular video of $T$ frames, our first step toward 4D reconstruction is recovering long-range 3D trajectories of scene points over time. Each trajectory is defined as a sequence $\{\boldsymbol{\tau}^t\}_{t=1}^T$, where $\boldsymbol{\tau}^t \in \mathbb{R}^3$ denotes the 3D position at frame $t$. However, estimating such trajectories from monocular inputs is inherently ill-posed due to depth ambiguity from 3D-to-2D projection. To mitigate this, we combine metric depth estimation [28, 29, 58] with long-range pixel tracking [12, 36, 73]. We predict per-frame depth maps $d^t : \mathbb{R}^2 \to \mathbb{R}_+$ and extract 2D trajectory $\{\mathbf{u}^t\}_{t=1}^T$, where $\mathbf{u}^t \in \mathbb{R}^2$ denotes the pixel location at frame $t$. Consistent with common practices [64, 73, 39], we then lift each 2D trajectory into 3D world space, aided by the depth priors:

$$\boldsymbol{\tau}^t = \mathbf{W}^t \pi_{\mathbf{K}}^{-1}\left(\mathbf{u}^t, d^t(\mathbf{u}^t)\right), \tag{2}$$

where $\pi_{\mathbf{K}}(\cdot)$ denotes the projection function from camera to image space defined by intrinsics $\mathbf{K}$, and $\mathbf{W}^t$ is the camera pose at frame $t$. When camera parameters are unavailable, we estimate them via bundle adjustment [45, 39].

The resulting sequence $\{\boldsymbol{\tau}^t\}_{t=1}^T$ provides a 3D trajectory across the scene. We further extract principal forward directions from these trajectories to construct oriented anchors, which provide structured and temporally coherent guidance for dynamic modeling in our framework.

## 4 Method

We introduce **Orientation-anchored Gaussian Splatting** (**OriGS**), a novel framework for 4D reconstruction from casual videos. OriGS is built upon two synergistic components: **(i)** a **Global Orientation Field** (Section 4.1) that captures long-range scene evolution via propagated orientation cues; and **(ii)** an **Orientation-aware Hyper-Gaussian** (Section 4.2) that models complex region-specific dynamics via orientation-conditioned inference in a unified hyperdimensional representation.

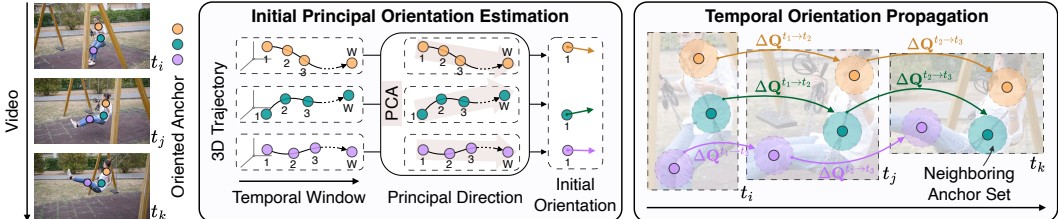

Figure 1: **Illustration of Global Orientation Field.** Given 3D trajectories recovered from monocular video, we extract a structured set of oriented anchors. **(Left)** Anchors' positions are initialized along tracked points across frames. **(Middle)** We estimate initial principal orientations by applying PCA over short temporal windows, reflecting the dominant forward direction per anchor. **(Right)** These initial orientations are then propagated over time via localized Procrustes alignment, producing a temporally coherent orientation field across the scene.

## 4.1 Global Orientation Field

**Initial Principal Orientation Estimation.** Given the recovered 3D trajectory $\{\boldsymbol{\tau}^t\}_{t=1}^T$ of scene points (Section 3), we convert these raw translation paths into a structured set of *oriented anchors*, each carrying a spatial position and an associated local orientation. These anchors serve as the basis for constructing our Global Orientation Field.

To initialize orientation, we estimate the principal direction of each anchor by analyzing the dominant motion in the early portion of its trajectory. Concretely, we unfold each trajectory and extract the first $W$ frames as a temporal window, forming $\{\boldsymbol{\tau}^t\}_{t=1}^W$. We center the points using the temporal mean $\bar{\boldsymbol{\tau}} = \frac{1}{W} \sum_{t=1}^W \boldsymbol{\tau}^t$, and define $\hat{\boldsymbol{\tau}}^t = \boldsymbol{\tau}^t - \bar{\boldsymbol{\tau}}$. We then compute the covariance matrix of the centered points and apply principal component analysis (PCA) to extract the dominant direction:

$$\text{PCA}(\mathbf{C}) = [\mathbf{v}_1, \mathbf{v}_2, \mathbf{v}_3], \quad \text{where} \quad \mathbf{C} = \frac{1}{W} \sum_{t=1}^W \hat{\boldsymbol{\tau}}^t \hat{\boldsymbol{\tau}}^{t\top}. \tag{3}$$

The leading eigenvector $\mathbf{v}_1$ reflects the dominant motion axis, which we assign as the anchor's forward direction to construct its initial local orientation $\mathbf{O}^1 \in SO(3)$.

**Temporal Orientation Propagation.** To extend the initial principal orientation across time, we estimate a time-varying orientation sequence $\{\mathbf{O}^t\}_{t=1}^T$ for each anchor, capturing its evolving motion direction throughout the scene. This can be formulated as a Procrustes alignment problem [26, 25]. At each time step $t$, we identify a local neighborhood $\{\boldsymbol{\tau}_k^t\}_{k=1}^K$ around each anchor based on Euclidean distance and align it to its reference configuration by optimizing:

$$\min_{\mathbf{T}^t \in SO(3)} \sum_{k=1}^K \left\| (\boldsymbol{\tau}_k^t - \bar{\boldsymbol{\tau}}^t) - \mathbf{T}^t (\boldsymbol{\tau}_k^1 - \bar{\boldsymbol{\tau}}^1) \right\|^2, \tag{4}$$

where $\bar{\boldsymbol{\tau}}^t$ and $\bar{\boldsymbol{\tau}}^1$ are spatial centroids of the anchor set, and $\mathbf{T}^t$ represents the rotation transformation.

This alignment is solved by applying singular value decomposition (SVD) to the covariance matrix of the localized anchor set. The optimal rotation transformation is then given by $\mathbf{T}^{t*} = \mathbf{V}\mathbf{U}^\top$, where $\mathbf{V}$ and $\mathbf{U}$ are obtained from SVD. As a correction step, we adjust the last column based on the sign of its determinant. Finally, we compute the anchor's orientation $\mathbf{O}^t$ by composing the optimal transformation with the initial principal orientation:

$$\mathbf{O}^t = \mathbf{T}^{t*} \cdot \mathbf{O}^1. \tag{5}$$

Repeating this process yields temporally coherent orientations $\{\mathbf{O}^t\}_{t=1}^T$. These propagated orientations constitute our Global Orientation Field, which encodes motion tendencies across the scene and provides stable anchors for modeling complex dynamics. We illustrate this process in Figure 1, where $\Delta\mathbf{Q}_i^{t \to t'} = (\mathbf{O}_i^{t'}, \boldsymbol{\tau}_i^{t'})(\mathbf{O}_i^t, \boldsymbol{\tau}_i^t)^{-1}$ denotes the relative anchor transformation in $SE(3)$ across time.

## 4.2 Orientation-aware Hyper-Gaussian

Dynamic scenes often exhibit complex, region-specific motion that cannot be fully explained by position or geometry alone. While casual monocular videos lack multi-view supervision, they retain

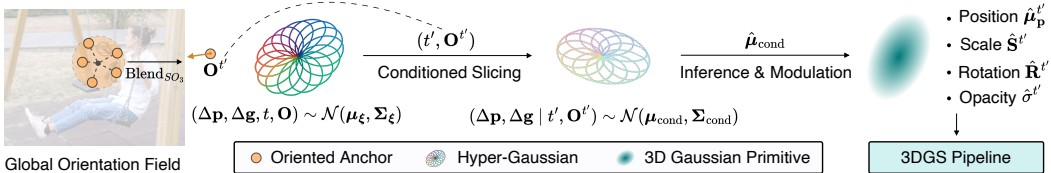

Figure 2: **Illustration of Orientation-aware Hyper-Gaussian.** We model complex dynamics using high-dimensional Gaussians defined over a structured manifold spanning time, space, geometry, and orientation. Given a target time $t'$ and its local orientation $\mathbf{O}^{t'}$ (interpolated from the Global Orientation Field via $SO(3)$ blending), we perform conditioned slicing on the hyper-Gaussian to infer the expected dynamic states. The predicted $\hat{\boldsymbol{\mu}}_{\text{cond}}$ modulates 3D Gaussian primitives, which is then compatible with the 3DGS pipeline.

coherent orientation cues that implicitly reveal how different regions are expected to move. We leverage this structural signal and introduce **Orientation-aware Hyper-Gaussian**, a hyperdimensional representation that models local dynamics over a unified manifold spanning time, space, geometry, and orientation. By conditioning on the evolving orientation field, this formulation facilitates principled deformation inference aligned with the dynamic intent.

**Multivariate Modeling of Dynamic States.** To reflect complex motion patterns across space and time, we represent each region of the scene with a multivariate dynamic state that jointly encodes temporal progression and structural changes. Specifically, let $\mathbf{p}^t \in \mathbb{R}^3$ denote the position of a point at time $t$. We define its local dynamic state as:

$$\boldsymbol{\xi} = (\Delta\mathbf{p}, \Delta\mathbf{g}, t, \mathbf{O}) \in \mathcal{M}, \tag{6}$$

where $\Delta\mathbf{p}$ and $\Delta\mathbf{g}$ represent deviations in position and geometry (*e.g.*, scale and rotation), $\mathbf{O} \in SO(3)$ is the orientation derived from our Global Orientation Field, and $\mathcal{M}$ denotes a manifold spanning time, space, geometry, and orientation.

This formulation captures the intrinsic interdependence among these factors in dynamic scenes, treating them as jointly evolving rather than isolated variables. To model their correlations in a unified and flexible manner, we then represent each dynamic state $\boldsymbol{\xi}$ as a probabilistic entity with a multivariate Gaussian over $\mathcal{M}$:

$$\boldsymbol{\xi} \sim \mathcal{N}(\boldsymbol{\mu_\xi}, \boldsymbol{\Sigma_\xi}), \quad \boldsymbol{\mu_\xi} \in \mathcal{M}, \quad \boldsymbol{\Sigma_\xi} \in \mathbb{R}^{D \times D}, \tag{7}$$

where $\boldsymbol{\mu_\xi}$ is the canonical dynamic state, and $\boldsymbol{\Sigma_\xi}$ captures intra- and inter-correlations. By grounding this probabilistic state in the evolving global orientation field, our model is able to learn region-specific variability that aligns with the scene's underlying dynamic intent.

**Hyper-Gaussian Representation.** To realize our formulation, we build upon 3D Gaussian Splatting [37] and equip each primitive with the multivariate Gaussian distribution defined in Eq. (7). This results in a dynamic representation that encodes the localized motion and geometric variability within a unified probabilistic entity.

Specifically, we begin by initializing a set of 3D Gaussians $\{\mathcal{G}_j\}$ across the scene, each parameterized by its position $\boldsymbol{\mu}_{\mathbf{P}j}$, rotation $\mathbf{R}_j$, scale $\mathbf{S}_j$, opacity $\sigma_j$, and color $\mathbf{c}_j$ (Section 3). To enable dynamic modeling, we associate each $\mathcal{G}_j$ with a state mean $\boldsymbol{\mu_\xi}$ and covariance $\boldsymbol{\Sigma_\xi}$:

$$\boldsymbol{\mu_\xi} = (\boldsymbol{\mu}_{\Delta\mathbf{p}}, \boldsymbol{\mu}_{\Delta\mathbf{g}}, \mu_t, \boldsymbol{\mu_\mathbf{O}}), \quad \boldsymbol{\Sigma_\xi} = \begin{bmatrix} \boldsymbol{\Sigma}_{(\Delta\mathbf{p}, \Delta\mathbf{g})} & \boldsymbol{\Sigma}_{(\Delta\mathbf{p}, \Delta\mathbf{g}),(t, \mathbf{O})} \\ \boldsymbol{\Sigma}^\top_{(\Delta\mathbf{p}, \Delta\mathbf{g}),(t, \mathbf{O})} & \boldsymbol{\Sigma}_{(t, \mathbf{O})} \end{bmatrix}, \tag{8}$$

where $\boldsymbol{\mu}_{\Delta\mathbf{p}} \in \mathbb{R}^3$ represents the spatial offset, $\boldsymbol{\mu}_{\Delta\mathbf{g}} \in \mathbb{R}^3_+ \times SO(3)$ describes local variations in scale and rotation, $\mu_t \in \mathbb{R}$ is the temporal coordinate, and $\boldsymbol{\mu_\mathbf{O}} \in SO(3)$ represents the orientation within the global field. The covariance matrix captures both intra- (*e.g.,* temporal or geometric) and cross-group (*e.g.,* spatial-orientational) dependencies to model complex dynamics.

For numerical stability, we parameterize $\boldsymbol{\Sigma_\xi}$ via Cholesky decomposition [11, 22, 23]: $\boldsymbol{\Sigma_\xi} = \mathbf{L_\xi}\mathbf{L_\xi}^\top$, where $\mathbf{L_\xi}$ is a lower-triangular matrix. Both $\boldsymbol{\mu_\xi}$ and $\mathbf{L_\xi}$ are treated as learnable parameters and jointly optimized with other Gaussian attributes. More details are provided in the supplementary materials.

**Conditioned Slicing for Dynamic State Inference.** Given the hyper-Gaussian representation, we seek to infer concrete dynamic behavior by conditioning on local motion context. Our core intuition

is that each part of the scene undergoes diverse localized deformation in response to how the region is expected to evolve under temporal and orientational cues.

Specifically, we define a query $(t', \mathbf{O}^{t'}) \in \mathbb{R} \times SO(3)$, representing the target time and its associated orientation. Given the probability $P(\boldsymbol{\xi}) = \mathcal{N}(\boldsymbol{\mu_\xi}, \boldsymbol{\Sigma_\xi})$, we derive the conditional distribution over the spatial and geometric subspace:

$$P(\Delta\mathbf{p}, \Delta\mathbf{g} \mid t', \mathbf{O}^{t'}) = \mathcal{N}(\boldsymbol{\mu}_{\text{cond}}, \boldsymbol{\Sigma}_{\text{cond}}), \tag{9}$$

with conditional mean and covariance given by Gaussian slicing rules:

$$\boldsymbol{\mu}_{\text{cond}} = (\boldsymbol{\mu}_{\Delta\mathbf{p}}, \boldsymbol{\mu}_{\Delta\mathbf{g}}) + \boldsymbol{\Sigma}_{(\Delta\mathbf{p},\Delta\mathbf{g}),(t,\mathbf{O})} \boldsymbol{\Sigma}_{(t,\mathbf{O})}^{-1} (t' - \mu_t, \mathbf{O}^{t'} \ominus \boldsymbol{\mu}_\mathbf{O}), \tag{10}$$

$$\boldsymbol{\Sigma}_{\text{cond}} = \boldsymbol{\Sigma}_{(\Delta\mathbf{p},\Delta\mathbf{g})} - \boldsymbol{\Sigma}_{(\Delta\mathbf{p},\Delta\mathbf{g}),(t,\mathbf{O})} \boldsymbol{\Sigma}_{(t,\mathbf{O})}^{-1} \boldsymbol{\Sigma}_{(t,\mathbf{O}),(\Delta\mathbf{p},\Delta\mathbf{g})}, \tag{11}$$

where $\ominus$ denotes the relative rotation operator in $SO(3)$. Practically, rather than sampling $(\Delta\mathbf{p}, \Delta\mathbf{g})$ from the conditional distribution, which may introduce optimization instability, we adopt the maximum-likelihood estimation principle and use the conditional mean $\boldsymbol{\mu}_{\text{cond}}$ as the deterministic prediction:

$$\hat{\boldsymbol{\mu}}_{\text{cond}} := \mathbb{E}[\Delta\mathbf{p}, \Delta\mathbf{g} \mid t', \mathbf{O}^{t'}] = \boldsymbol{\mu}_{\text{cond}}. \tag{12}$$

This inferred local dynamic state captures the expected position and geometry variations given temporal and orientation contexts. Through this slicing process, the hyper-Gaussian can be adaptively projected onto various scene regions, enabling temporally and spatially coherent reconstruction.

**Modulation of Hyper-Gaussian Attributes.** To integrate our dynamic modeling into the efficient and differentiable rendering pipeline, we modulate 3D Gaussian primitives based on the inferred dynamic state. For a target time $t'$ and local orientation $\mathbf{O}^{t'}$, we can obtain $\hat{\boldsymbol{\mu}}_{\text{cond}}$ from the conditioned distribution using Eq. (12) and apply it to the 3D Gaussian parameters accordingly:

$$\hat{\boldsymbol{\mu}}_\mathbf{P}^{t'} = \boldsymbol{\mu}_\mathbf{P}^{t'} + \hat{\boldsymbol{\mu}}_{\Delta\mathbf{p}}, \quad \hat{\mathbf{S}}^{t'} = \mathbf{S} + \hat{\boldsymbol{\mu}}_{\Delta\mathbf{S}}, \quad \hat{\mathbf{R}}^{t'} = \mathbf{R}^{t'} \oplus \hat{\boldsymbol{\mu}}_{\Delta\mathbf{R}}, \tag{13}$$

where $\hat{\boldsymbol{\mu}}_{\Delta\mathbf{p}}, \hat{\boldsymbol{\mu}}_{\Delta\mathbf{S}}, \hat{\boldsymbol{\mu}}_{\Delta\mathbf{R}}$ denote the inferred variations in position, scale, and rotation, respectively, and $\oplus$ is quaternion-based rotation composition. In parallel, we modulate each primitive's opacity based on its proximity to the canonical state in both time and orientation. This confidence-weighted visibility is defined as:

$$\hat{\sigma}^{t'} = \sigma^{t'} \cdot \exp\left(-\frac{1}{2}\left[\Sigma_t^{-1}(t' - \mu_t)^2 + \left(\mathbf{O}^{t'} \ominus \boldsymbol{\mu}_\mathbf{O}\right)^\top \boldsymbol{\Sigma}_\mathbf{O}^{-1} \left(\mathbf{O}^{t'} \ominus \boldsymbol{\mu}_\mathbf{O}\right)\right]\right). \tag{14}$$

Together, these modulations encourage each hyper-Gaussian to adaptively express geometry and appearance consistent with the temporal and orientational cues, while suppressing noisy contributions from out-of-support regions.

**Anchor-Guided Deformation and Conditioning.** Recall that to query the hyper-Gaussian via conditioned slicing defined by Eq. (9), we must first estimate each Gaussian's local motion context, including its position and orientation under the evolving scene. We achieve this through anchor-driven deformation within our Global Orientation Field, drawing inspiration from Embedded Deformation Graphs [65, 69] and their extensions in dynamic modeling [5, 13, 44, 39, 52, 83].

Specifically, each Gaussian $\mathcal{G}_j$ is softly associated with a set of $K$ nearby oriented anchors via skinning weights $\{w_{ij}\}$ computed from spatial proximity. Each anchor $i$ carries a time-varying pair $(\mathbf{O}_i^t, \boldsymbol{\tau}_i^t)$, representing its orientation and position at time $t$. We express the relative anchor transformation in $SE(3)$ across time as: $\Delta\mathbf{Q}_i^{1\to t'} = (\mathbf{O}_i^{t'}, \boldsymbol{\tau}_i^{t'})(\mathbf{O}_i^1, \boldsymbol{\tau}_i^1)^{-1}$, which is further converted into a dual quaternion $\mathbf{q}_i^{1\to t'} \in \mathbb{D}$. To deform $\mathcal{G}_j$ towards any target time $t'$, we aggregate the relative transformations of its neighboring anchors using a weighted dual quaternion blend [52]: $\hat{\mathbf{q}}_j^{1\to t'} = \text{Blend}_\mathbb{D}\left(\{w_{ij}, \mathbf{q}_i^{1\to t'}\}_{i=1}^K\right)$, which then yields the updated attributes:

$$(\mathbf{R}_j^{t'}, \boldsymbol{\mu}_{\mathbf{P}j}^{t'}) = \hat{\mathbf{q}}_j^{1\to t'}(\mathbf{R}_j^1, \boldsymbol{\mu}_{\mathbf{P}j}^1). \tag{15}$$

Importantly, this deformation scheme provides access to the local orientation $\mathbf{O}^{t'}$ required for state conditioning in Eq. (9). However, this requires additional care since the deformed Gaussian may not coincide with any specific anchor. Therefore, we interpolate $\mathbf{O}^{t'}$ from surrounding anchors:

$$\mathbf{O}^{t'} = \text{Blend}_{SO(3)}\left(\{w_{ij}, \mathbf{O}_i^{t'}\}_{i=1}^K\right). \tag{16}$$

| MoSca [39] (Training View) | **OriGS** (Ours) (Training View) | MoSca [39] (Novel View) | **OriGS** (Ours) (Novel View) |

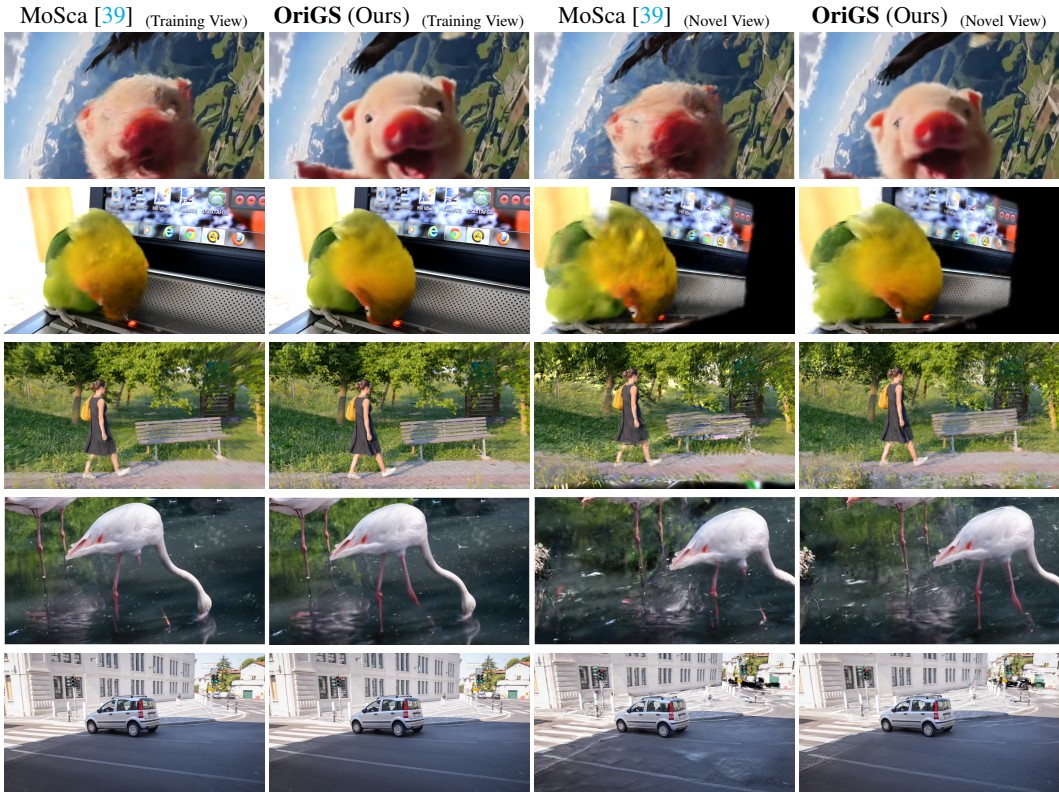

Figure 3: **Visual comparisons of training sequence reconstruction and novel view synthesis on in-the-wild videos.** Please 🔍 zoom in for more details.

This produces a smoothly varying orientation field that respects both the global structure and local dynamics. By grounding the conditioned slicing on these anchor-driven estimates, our hyper-Gaussian representation can adaptively capture complex motion across different regions.

## 5 Experiments

### 5.1 Experimental Protocol

**Datasets.** We evaluate the 4D reconstruction and novel view synthesis performance of our OriGS on a diverse set of in-the-wild monocular videos, emphasizing scenes with complex motion and object interactions. Our primary evaluation includes videos from DAVIS [59], OpenAI SORA [6], and YouTube-VOS [74], which reflect the casual and unconstrained nature of our target setting. For quantitative benchmarking, we additionally report results on the DyCheck dataset [19], which contains seven scenes with multi-camera captures for novel view synthesis evaluation.

**Evaluation.** For in-the-wild videos, where ground-truth novel views are inherently unavailable, we conduct qualitative evaluations to assess the visual fidelity of reconstructed scenes. For the DyCheck dataset [19], which provides multi-camera setups for quantitative assessment, we evaluate our method using standard image quality metrics: Peak Signal-to-Noise Ratio (PSNR), Structural Similarity Index Measure (SSIM) [68], and Learned Perceptual Image Patch Similarity (LPIPS) [78]. The DyCheck dataset uses three synchronized cameras: one hand-held mobile view and two stationary references. Following [19], we measure reconstruction quality from the reference viewpoints.

### 5.2 Experimental Results

**In-the-wild.** We qualitatively evaluate OriGS on a range of casually captured monocular videos that exhibit diverse motion patterns. Figure 3 presents side-by-side comparisons with MoSca [39], a recent state-of-the-art system for automatic 4D reconstruction in real-world scenarios. Across

Table 1: **Quantitative comparisons of novel view synthesis on DyCheck.** We report PSNR, SSIM, and LPIPS metrics across seven scenes. Markers ○ and ● denote with and without ground-truth camera pose, respectively. The best , second best , and third best results are highlighted.

| Method | Apple PSNR↑ | SSIM↑ | LPIPS↓ | Block PSNR↑ | SSIM↑ | LPIPS↓ | Paper Windmill PSNR↑ | SSIM↑ | LPIPS↓ | Space Out PSNR↑ | SSIM↑ | LPIPS↓ |
|---|---|---|---|---|---|---|---|---|---|---|---|---|
| ○ T-NeRF [19] | 17.43 | 0.728 | 0.508 | 17.52 | 0.669 | 0.346 | 17.55 | 0.367 | 0.258 | 17.71 | 0.591 | 0.377 |
| ○ NSFF [42] | 16.47 | 0.754 | 0.414 | 14.71 | 0.606 | 0.438 | 14.94 | 0.272 | 0.348 | 17.65 | 0.636 | 0.341 |
| ○ Nerfies [55] | 17.54 | 0.750 | 0.478 | 16.61 | 0.639 | 0.389 | 17.34 | 0.378 | 0.211 | 17.79 | 0.622 | 0.303 |
| ○ HyperNeRF [56] | 17.64 | 0.743 | 0.478 | 17.54 | 0.670 | 0.331 | 17.38 | 0.382 | 0.209 | 17.93 | 0.605 | 0.320 |
| ● RoDynRF [47] | 18.73 | 0.722 | 0.552 | 18.73 | 0.634 | 0.513 | 16.71 | 0.321 | 0.482 | 18.56 | 0.594 | 0.413 |
| ○ DynPoint [80] | 17.78 | 0.743 | - | 17.67 | 0.667 | - | 17.32 | 0.366 | - | 17.78 | 0.603 | - |
| ○ PGDVS [79] | 16.66 | 0.721 | 0.411 | 16.38 | 0.601 | 0.293 | 17.19 | 0.386 | 0.277 | 16.49 | 0.592 | 0.326 |
| ○ DyBluRF [7] | 18.00 | 0.737 | 0.488 | 17.47 | 0.665 | 0.349 | 18.19 | 0.405 | 0.301 | 18.83 | 0.643 | 0.326 |
| ○ CTNeRF [50] | 19.53 | 0.691 | - | 19.74 | 0.626 | - | 17.66 | 0.346 | 0.346 | 18.11 | 0.601 | - |
| ○ Dyn. GS [49] | 7.65 | - | 0.766 | 7.55 | - | 0.684 | 6.24 | - | 0.729 | 6.79 | - | 0.733 |
| ○ 4DGS [71] | 15.41 | - | 0.450 | 11.28 | - | 0.633 | 15.60 | - | 0.297 | 14.60 | - | 0.372 |
| ○ D-NPC [35] | 16.83 | 0.752 | 0.469 | 15.53 | 0.632 | 0.350 | 18.01 | 0.432 | 0.209 | 18.69 | 0.640 | 0.246 |
| ● Marbles [64] | 16.50 | - | 0.499 | 16.11 | - | 0.363 | 16.19 | - | 0.454 | 15.97 | - | 0.437 |
| ○ SoM [67] | - | - | - | - | - | - | - | - | - | - | - | - |
| ● MoSca [39] | 15.99 | 0.705 | 0.503 | 18.20 | 0.665 | 0.324 | 21.37 | 0.645 | 0.175 | 22.57 | 0.750 | 0.198 |
| ● OriGS (Ours) | 17.21 | 0.739 | 0.428 | 18.67 | 0.676 | 0.317 | 21.48 | 0.646 | 0.171 | 22.78 | 0.754 | 0.194 |
| ○ OriGS (Ours) | 19.46 | 0.807 | 0.341 | 18.76 | 0.690 | 0.319 | 22.46 | 0.751 | 0.152 | 20.87 | 0.672 | 0.249 |

| Method | Spin PSNR↑ | SSIM↑ | LPIPS↓ | Teddy PSNR↑ | SSIM↑ | LPIPS↓ | Wheel PSNR↑ | SSIM↑ | LPIPS↓ | **Average** PSNR↑ | SSIM↑ | LPIPS↓ |
|---|---|---|---|---|---|---|---|---|---|---|---|---|
| ○ T-NeRF [19] | 19.16 | 0.567 | 0.443 | 13.71 | 0.570 | 0.429 | 15.65 | 0.548 | 0.292 | 16.96 | 0.577 | 0.379 |
| ○ NSFF [42] | 17.26 | 0.540 | 0.371 | 12.59 | 0.537 | 0.527 | 14.59 | 0.511 | 0.331 | 15.46 | 0.551 | 0.396 |
| ○ Nerfies [55] | 18.38 | 0.585 | 0.309 | 13.65 | 0.557 | 0.372 | 13.82 | 0.458 | 0.310 | 16.45 | 0.570 | 0.339 |
| ○ HyperNeRF [56] | 19.20 | 0.561 | 0.325 | 13.97 | 0.568 | 0.350 | 13.99 | 0.455 | 0.310 | 16.81 | 0.569 | 0.332 |
| ● RoDynRF [47] | 17.41 | 0.484 | 0.570 | 14.33 | 0.536 | 0.613 | 15.20 | 0.449 | 0.478 | 17.10 | 0.534 | 0.517 |
| ○ DynPoint [80] | 19.04 | 0.564 | - | 13.95 | 0.551 | - | 14.72 | 0.515 | - | 16.89 | 0.573 | - |
| ○ PGDVS [79] | 18.49 | 0.590 | 0.247 | 13.29 | 0.516 | 0.399 | 12.68 | 0.429 | 0.429 | 15.88 | 0.548 | 0.340 |
| ○ DyBluRF [7] | 18.20 | 0.541 | 0.400 | 14.61 | 0.572 | 0.435 | 16.26 | 0.575 | 0.325 | 17.37 | 0.591 | 0.373 |
| ○ CTNeRF [50] | 19.79 | 0.516 | - | 14.51 | 0.509 | - | 14.48 | 0.430 | - | 17.69 | 0.531 | - |
| ○ Dyn. GS [49] | 8.08 | - | 0.651 | 7.41 | - | 0.690 | 7.28 | - | 0.593 | 7.29 | - | 0.692 |
| ○ 4DGS [71] | 14.42 | - | 0.339 | 12.36 | - | 0.466 | 11.79 | - | 0.436 | 13.64 | - | 0.428 |
| ○ D-NPC [35] | 17.78 | 0.585 | 0.309 | 12.19 | 0.536 | 0.503 | 13.27 | 0.549 | 0.349 | 16.04 | 0.589 | 0.348 |
| ● Marbles [64] | 17.51 | - | 0.424 | 13.68 | - | 0.443 | 14.58 | - | 0.389 | 15.79 | - | 0.428 |
| ○ SoM [67] | - | - | - | - | - | - | - | - | - | 17.32 | 0.598 | 0.296 |
| ● MoSca [39] | 20.71 | 0.677 | 0.229 | 15.54 | 0.624 | 0.356 | 17.74 | 0.667 | 0.238 | 18.84 | 0.676 | 0.289 |
| ● OriGS (Ours) | 20.92 | 0.670 | 0.215 | 16.11 | 0.639 | 0.334 | 17.92 | 0.670 | 0.233 | 19.30 | 0.685 | 0.270 |
| ○ OriGS (Ours) | 21.62 | 0.759 | 0.177 | 16.45 | 0.649 | 0.325 | 18.19 | 0.684 | 0.226 | 19.69 | 0.716 | 0.256 |

both training sequence reconstruction and novel view synthesis, OriGS consistently recovers cleaner geometry and preserves structural integrity under complex dynamics. Notably, our method remains robust in challenging scenarios, such as fast movement (*e.g.*, the spinning pig and the parrot). These results demonstrate that OriGS generalizes effectively to complex real-world videos, enabling faithful 4D reconstruction and compelling view synthesis without relying on multi-view constraints.

**DyCheck.** We compare OriGS against recent state-of-the-art methods, including NeRF-based [19, 42, 55, 56, 47, 7, 50, 80, 79] and 3DGS-based [49, 71, 35, 64, 67, 39]. Table 1 reports quantitative results across all seven scenes using PSNR, SSIM, and LPIPS. We indicate whether ground-truth camera poses are used by markers ○ (with) and ● (without). Our OriGS achieves superior reconstruction fidelity, especially in scenes involving rotational deformation (*e.g.*, the "Apple" and "Paper Windmill" scenes), where prior methods tend to produce artifacts. Notably, OriGS outperforms all baselines in the average across all three metrics, even without access to ground-truth camera pose. When provided with accurate poses, performance is further improved in most scenes, demonstrating the effectiveness of our framework. A minor exception is the "Space Out" scene, which is limited by inaccurate pose metadata provided in the dataset, as previously observed in [67].

Figure 4 showcases the qualitative advantages of OriGS in novel view synthesis. Compared to mainstream methods, our OriGS can produce sharper surface boundaries and more coherent geometry under challenging motion. For example, in the "Paper Windmill" scene (top row), other methods typically struggle with blurred or broken structures due to the rapid spinning motion. In contrast, our OriGS, by anchoring complex dynamics to a coherent orientation field, maintains the integrity of the rotating blades. Compared to OriGS, baseline results often exhibit severe ghosting artifacts and

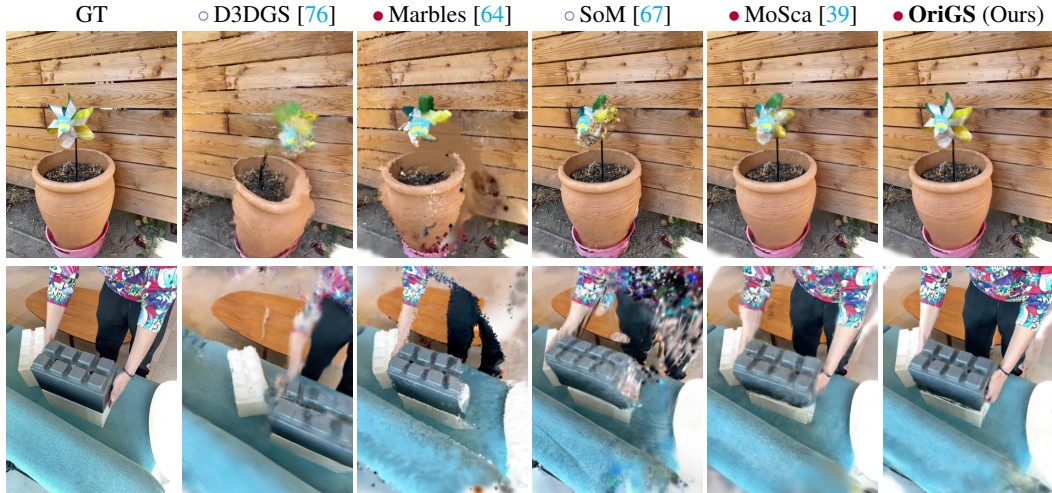

| GT | ○ D3DGS [76] | ● Marbles [64] | ○ SoM [67] | ● MoSca [39] | ● **OriGS** (Ours) |

Figure 4: **Visual comparisons of novel view synthesis on DyCheck.** Markers ○ and ● denote with and without ground-truth camera pose, respectively. Our OriGS can recover sharper geometry and more coherent motion.

rigid-body distortion in scenes with hand-object occlusion (*e.g.*, the "Block" scene (bottom row)), further emphasizing the superiority of our method in handling complex motion.

## 5.3    Ablation Study

To analyze the impact of each design component in OriGS, we conduct a progressive ablation study from a minimal baseline to our full model. Table 2 reports quantitative results on the "Apple" scene from DyCheck [19] using the provided camera pose, and Figure 5 presents qualitative comparisons on the "Libby" scene from DAVIS [59]. We follow the same experimental setting for novel view synthesis as in Section 5.2. **(i)** We begin with **3DGS-MLP** as the baseline, where each 3D Gaussian primitive is updated individually across time using a shared deformation MLP, without explicit motion modeling.

Table 2: **Ablation study on OriGS variants on the "Apple" scene from DyCheck [19].**

| Method | PSNR↑ | SSIM↑ | LPIPS↓ |
|---|---|---|---|
| (i) 3DGS-MLP (Baseline) | 13.47 | 0.532 | 0.586 |
| (ii) Deform w/ GOF | 16.28 | 0.694 | 0.476 |
| (iii) Hyper-Gaussian w/ $t$ | 18.71 | 0.750 | 0.393 |
| (iv) **OriGS** (Full) | **19.46** | **0.807** | **0.341** |

This baseline can only capture simple movement and struggles with structural coherence. **(ii)** Next, we consider **Deform w/ GOF**, which replaces the MLP with anchor-driven deformation guided by our Global Orientation Field. This variant enables Gaussian primitives to capture global motion, yet lacks the expressiveness to model complex dynamics across regions. **(iii)** To enable localized, time-adaptive behavior, we augment the model with **Hyper-Gaussian w/ $t$**, where each Gaussian encodes a probabilistic state over time, space, and geometry (*i.e.*, $\xi = (\Delta\mathbf{p}, \Delta\mathbf{g}, t)$). The deformation is inferred by conditioning this hyper-Gaussian on the target time step, enabling smoother and temporally adaptive reconstruction. **(iv)** Finally, our **OriGS** further incorporates orientation as a conditioning signal. By anchoring local dynamics to both temporal and orientational cues, it enables the model to capture direction-sensitive motion patterns and better preserve coherent evolution across regions with diverse dynamics.

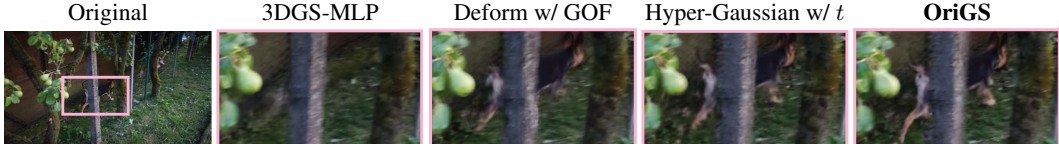

| Original | 3DGS-MLP | Deform w/ GOF | Hyper-Gaussian w/ $t$ | **OriGS** |

Figure 5: **Ablation study on OriGS variants on the "Libby" scene from DAVIS [59].** We show a frame from the original input video (leftmost) and zoomed-in novel view synthesis results from different ablated variants. The main region of interest is  the dog behind the pillar , which exhibits fast movement and heavy occlusion.

# 6 Conclusion

In this work, we presented Orientation-anchored Gaussian Splatting (OriGS), a novel framework for 4D reconstruction from casually captured monocular videos. Our core insight is to embrace orientation as a dynamic anchor that organizes how different parts of the scene deform and interact. By introducing a Global Orientation Field and a Hyper-Gaussian representation, OriGS models long-range evolution and local motion variation in a unified and principled manner, thereby capturing coherent dynamics across the scene. Extensive experiments demonstrate the superiority of our OriGS, enabling high-fidelity reconstruction and view synthesis under challenging real-world scenarios.

**Acknowledgments:** This research is supported by NSF IIS-2525840, CNS-2432534, ECCS-2514574, NIH 1RF1MH133764-01 and Cisco Research unrestricted gift. This article solely reflects opinions and conclusions of authors and not funding agencies.

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

## A Evaluation on Point Tracking

Following the DyCheck benchmark [19] and prior works [64, 39], we perform a quantitative evaluation on point tracking accuracy. We compare with both tracking and reconstruction methods. Results shown in Table 3 indicate the superiority of our OriGS.

Table 3: Correspondence Evaluation on DyCheck with PCK-T @0.05%.

| Method | Nerfies [55] | HyperNeRF [56] | Dyn. GS [49] | 4DGS [71] |
|---|---|---|---|---|
| PCK-T ↑ | 0.400 | 0.453 | 0.079 | 0.073 |
| Method | CoTracker [36] | Marbles [64] | MoSca [39] | **OriGS** (Ours) |
| PCK-T ↑ | 0.803 | 0.806 | 0.824 | **0.851** |

## B Demo Videos

To further showcase the effectiveness of OriGS in real-world scenarios, we include additional video demonstrations of in-the-wild 4D reconstruction in the supplementary material. Following the same experimental setup as in the main paper, we present visualizations of both training sequence reconstruction and novel view synthesis, comparing OriGS with the recent state-of-the-art 4D reconstruction system. In addition, we provide video demonstrations of the OriGS variants designed in our ablation study, illustrating how individual components affect reconstruction quality. These results collectively highlight the temporal consistency and structural fidelity of OriGS across diverse scenes and motion patterns.

## C Implementation Details

**Oriented Anchor Initialization.** We follow a similar initialization pipeline provided in recent open-source codebases of dynamic reconstruction frameworks [64, 39]. Specifically, we first extract long-range 2D point trajectories using SpatialTracker [73] and obtain per-frame depth maps using DepthCrafter [29]. These 2D correspondences are lifted into 3D space using estimated camera parameters from bundle adjustment [39]. The resulting 3D trajectories are then converted into oriented anchors, whose initial principal directions are estimated by applying PCA over early-frame motion. In our experiments, we set the temporal window size $W = 5$. This step provides a stable estimate of forward direction, which is then temporally propagated to form Global Orientation Field.

**Hyper-Gaussian Parameterization.** We adopt the 3D Gaussian Splatting (3DGS) [37] as the base representation and extend each Gaussian primitive with a hyper-Gaussian parameterized by a canonical state $\mu_{\xi}$ and a Cholesky-decomposed covariance $\Sigma_{\xi} = L_{\xi} L_{\xi}^{\top}$, where $L_{\xi}$ is a lower-triangular matrix [22, 23, 11, 24]. To improve computational efficiency, we avoid constructing the full covariance matrix $\Sigma_{\xi}$. Instead, we further factor it into two learnable subcomponents: **(i)** a marginal covariance $\Sigma_{(t,O)}$ over temporal and orientational variables, which inherits the Cholesky-decomposed parameterization to ensure positive semi-definiteness, and **(ii)** a cross-covariance term $\Sigma_{(\Delta p, \Delta g),(t,O)}$ that captures the correlation between dynamic geometry $(\Delta p, \Delta g)$ and temporal-orientational context. During training, we jointly optimize these covariance terms along with the canonical mean $\mu_{\xi}$ and the original 3DGS parameters (position, scale, rotation, color, and opacity). This factorization strategy circumvents the overhead of full matrix parameterization while retaining the necessary statistical coupling for efficient conditioned slicing.

**Optimization.** We optimize the model using the differentiable rasterization-based rendering pipeline from 3DGS [37], adapted to support high-dimensional slicing and dynamic modulation. The loss function combines: **(i)** photometric loss [37], an RGB reconstruction loss between rendered and ground-truth images, **(ii)** 2D correspondence loss, alignment to long-range 2D tracks and depth priors from foundation models, as in [64, 39, 44, 54], and **(iii)** deformation regularization, an as-rigid-as-possible constraint [73, 63, 1, 32] applied to anchor-guided transformations. For scalability and high-fidelity reconstruction, we also design a pruning-and-densification scheme: Gaussian primitives with low opacity are pruned, while spatial regions exhibiting high response gradients with respect to $\mu_p$ and $\mu_{\Delta p}$ are densified through local duplication. All experiments are conducted on a single

NVIDIA RTX A6000 GPU, and the full optimization of a typical scene takes approximately 0.5–2 hours, depending on video length and complexity.

## D    Limitations

While OriGS demonstrates promising results, it also presents certain limitations. **(i)** In scenes dominated by simple motion, such as near-rigid translation along a fixed axis, orientation tends to remain approximately constant over time. Consequently, conditioning on these orientational cues may offer limited additional benefit, yet our framework still incurs unnecessary optimization complexity of high-dimensional modeling. Future work could explore adaptive mechanisms that modulate the use or dimensionality of orientation cues based on the complexity of motion in the scene. **(ii)** OriGS leverages 2D priors such as point trajectories and depth from vision foundation models to initialize and optimize the orientation field. The reliability of these priors can affect the quality of 4D reconstruction. This reflects a deeper challenge tied to the development of reliable visual priors, which has long been a cornerstone of progress in computer vision research.

## E    Broader Impacts

OriGS provides a unified framework for reconstructing dynamic scenes from casual monocular videos, which can benefit various real-world applications. For instance, in virtual and augmented reality, OriGS can reconstruct dynamic environments or actors from consumer-grade video input alone, lowering the barrier to immersive content creation. Our work can also support robotics and behavioral analysis, especially where low-cost monocular capture is the only viable option. However, as with other scene reconstruction techniques, OriGS could potentially be misused for unauthorized replication of environments or individuals. While our framework is not designed for such misconduct, responsible usage should consider privacy and ethical concerns.

