# OpenReview forum: "Orientation-anchored Hyper-Gaussian for 4D Reconstruction from Casual Videos"
_NeurIPS.cc/2025/Conference — NeurIPS 2025 poster_

### Official Review · Reviewer_1JMD · 2025-06-13

**Clarity:** 3
**Significance:** 3
**Originality:** 4
**Rating:** 5
**Confidence:** 3

**Summary:**

This paper presents a novel gaussian splats optimization framework for dynamic 4D scene reconstruction from casually captured videos. This novel framework composes of two key parts:
- a method to estimate a global orientation field from consecutive frames using its metric depth estimation
- a probabilistic formulation of gaussian splats that conditions each guassian splat on the local orientation and time, which is able to capture more diverse and complex region-specific deformations through time.

**Questions:**

Mostly stated above. Among them, I care most about
- adding some form of visualization of the estimated global orientation field, or slices of it along with the results.

**Ethical Concerns:**

["NO or VERY MINOR ethics concerns only"]

**Final Justification:**

I have reviewed authors' responses and keeping my score.

**Limitations:**

Yes.

**Quality:**

4

**Strengths And Weaknesses:**

I really enjoyed reading the paper in general! It's well-written and very clear in its exposition. The method is straightforward and elegant with good results.

I only find the visualization a bit lacking given the insights the paper focuses on. The paper showed very little visualization of the actual orientation field and the conditioning/slicing mechanism with gaussian splats. Some form of this visualization that's not just abstract didactic ones could be quite nice to add to the paper in addition to the results.

In section 4.1 titled as Global Orientation Field, it doesn’t really mention how is the field interpolated in places where there is no anchor points estimated. It’s mentioned a bit later in line 241. I think it would really help with the paper flow if it’s explained in 4.1 how is the orientation field interpolated at places with no anchor points, as the gaussian splats can be placed anywhere in space.

Other minor things:
- line 210: would love to see some quantitative results on the difference between sampling from the distribution v.s using the expected mean ? Is one more expensive than the other?
- In the ablation study, I’m a bit confused on 3DGS + MLP v.s. Deform w/ GOF. Seems in the first case, there is a shared MLP being optimized, does the second one optimize anything to guide the deformation? Or it’s purely using the inferred gof to update splats? Could you provide more details of these ablated studies in supplement?

---

> ### Author Rebuttal · Authors · 2025-07-30
>
> We thank the reviewer for the kind comments and for highlighting important aspects to improve clarity and insight.
> # **W1: More Visualization**
> We appreciate the suggestion of visualizing the orientation field and the Gaussian slicing process, which would be meaningful. In the revised version, we plan to include the following visualizations:
>
> ### a. Global Orientation Field
> **(1) Anchor Tracking.** We will track oriented anchors and project them onto 2D to produce motion trajectories over time. Overlaying these trajectories on the RGB video frames offers an intuitive view of how the orientations follow the motion in dynamic scenes.
>
> **(2) Interpolated Orientation.** To visualize the interpolated orientation field, we will densely query orientations using deformed Gaussian positions. We then reduce these orientations to 3D-vector using PCA and apply a color-mapping scheme similar to optical flow to illustrate local directionality and temporal evolution.
>
> ### b. Slicing Mechanism
> **(1) 4D Primitive.** Using the Viser tool, we can directly visualize Gaussian primitives in the full 4D space, before and after applying the orientation-conditioned slicing. This helps understand how high-dimensional structures are sliced under different motion conditions.
>
> **(2) Rendering Quality.** We will also provide rendered image comparisons that show how the orientation-aware slicing affects the final visual quality.
>
> Unfortunately, due to **NeurIPS policies this year**, we are **NOT** permitted to include additional figures or visual materials during rebuttal. We sincerely regret this limitation, but we will ensure that the above visualizations are presented in the final supplementary materials.
>
>
> # **W2: Paper Flow Regarding Section 4.1**
> We sincerely appreciate the reviewer's constructive suggestion to improve the narrative structure of Section 4.1.
>
> **(1) Current Logit.** In the current draft, we **first** introduce the Hyper-Gaussian formulation and **only then** motivate the need to query arbitrary positions within the Global Orientation Field, which **in turn** leads to the introduction of the interpolation mechanism.
>
> **(2) Rethinking.** Upon revisiting this structure in light of the reviewer's comment, we realize that the **interpolation mechanism** is in fact more **foundational**, and not inherently tied to the Hyper-Gaussian slicing process. Rather, it is an **intrinsic property** of the Global Orientation Field, which by design should support querying at arbitrary 3D locations.
>
> **(3) Better Structure.** With this in mind, we agree that it would be more coherent to present the interpolation strategy directly within **Section 4.1**, as part of the core definition of the field. This adjustment will make the exposition more self-contained and improve the overall flow of the method section.
>
>
> # **W3: Sampling from the Distribution**
> We explored an alternative of our Hyper-Gaussian representation by sampling from the learned distribution, rather than conditioning on its mean.
>
> **(1) Implementation.** To ensure differentiability, we apply the reparameterization trick:
> $$z = \mu + \sigma \cdot \epsilon, \epsilon \sim \mathcal{N}(\mathbf{0}, \mathbf{I}). $$
> The deformation is then obtained using the sampled latent $z$ instead of the mean. We perform single-sample inference per primitive without Monte Carlo averaging.
>
> **(2) Performance Comparison.** We compare "sampling from the distribution" v.s. "using the expected mean" using the DyCheck benchmark present in Table 1 in our paper. We observe that the sampling variant results in inferior reconstruction quality compared to mean-based conditioning:
>
> | Methods             | PSNR↑ | SSIM↑ | LPIPS↓ |
> | - | - | - | - |
> | sampling | 18.93 | 0.621 | 0.271 |
> | mean | 19.69 | 0.716 | 0.256 |
>
> We hypothesize two reasons:
> - **Reason 1. 4D reconstruction from monocular video is fundamentally deterministic.** The learned mean of the Hyper-Gaussian corresponds to the most representative deformation mode, as it aligns with the highest-probability activation slice. Sampling may introduce unnecessary variance that disrupts the coherence of the deformation field and hinders stable optimization.
>
> - **Reason 2. Lack of meaningful uncertainty in monocular settings.** While stochastic sampling introduces diversity, the intrinsic ambiguity of monocular 4D reconstruction still remains. Handling such uncertainty would likely require a generative model capable of synthesizing the missing information from multiple views. Nonetheless, we believe this direction is orthogonal and complementary to our work, which focuses on building a robust 4D representation that could be integrated into such generative frameworks. We hope our contribution will benefit the broader goal of advancing 4D modeling from casual video.
>
> **(3) Efficiency.** From a computational standpoint, using a sampled latent vector incurs negligible overhead compared to using the mean, because both rely on a single forward pass through the same deformation query pipeline and the reparameterization operation itself is lightweight.
>
>
> # **W4: Clarification on the Ablation.**
> Thank you for pointing out this potential ambiguity. We will include a more detailed explanation in the supplementary material to improve clarity.
>
> **(1) 3DGS-MLP** introduces a shared MLP that is trained to predict per-primitive deformation.
>
> **(2) Deform w/ GOF**, as described in the main paper (line 300), removes the learnable MLP entirely. Instead, it purely uses the Global Orientation Field to directly drive deformation. Specifically, it adopts the Anchor-Guided Deformation detailed in lines 227–244, which shifts each Gaussian primitive along the oriented anchor trajectory. This variant allows Gaussian primitives to capture global motion trends, but lacks the expressiveness needed to model complex, region-specific dynamics.

---

> > ### Comment · Reviewer_1JMD · 2025-08-04
> >
> > Thank you to the authors for addressing my concerns -- I will keep my score.

---

> > > ### Author Response · Authors · 2025-08-04
> > > **Thank you**
> > >
> > > Dear Reviewer 1JMD,
> > >
> > > Thank you very much for your thoughtful feedback and engagement during the review process.
> > >
> > > Best regards,
> > >
> > > Authors

---

### Official Review · Reviewer_QZBh · 2025-07-03

**Clarity:** 1
**Significance:** 2
**Originality:** 2
**Rating:** 3
**Confidence:** 4

**Summary:**

This paper proposes a framework, called OriGS, for 4D reconstruction from monocular videos. The key to the proposed method is: 1) a Global Orientation Field, that estimates global forward moving directions, 2) an Orientation-aware Hyper-Gaussian representation, that queries local deformations by time and global orientations.

**Questions:**

Why are local deformations queried from global anchor orientations only, rather than from global anchor positions as well?

**Ethical Concerns:**

["NO or VERY MINOR ethics concerns only"]

**Limitations:**

Yes

**Quality:**

2

**Strengths And Weaknesses:**

Strength:
* Decomposition of motions into global and local components is reasonable.
* The proposed method surpasses existing baselines in quantitative evaluation.

Weakness:
* The advantages of the Orientation-aware Hyper-Gaussian representation is not clearly demonstrated. For example, in Fig. 5, I cannot the difference between the results w/ (col 4&5) and w/o (col 3) Hyper-Gaussian. The authors are suggested to highlight the difference in qualitative results and explain why different variants lead to such changes.
* The presentation of Hyper-Gaussian in Sec 4.2 is difficult to follow. I struggled a lot to figure out the design of this module.

---

> ### Author Rebuttal · Authors · 2025-07-30
>
> # **W1: Advantages of Orientation-aware Hyper-Gaussian**
> We appreciate the reviewer's suggestion to highlight the qualitative differences and analyze the behavior of these variants.
>
> ### **a. Visual Comparison**
> **(1) Difference.** In the "Libby" scene (Fig. 5), the most notable differences among the variants occur in **the running dog**, especially in highly articulated and fast-rotating parts such as the **legs, tail, and head**.
>
> **(2) Analysis on the "Libby" scene.** The "Libby" scene is particularly challenging due to both **fast motion** and **severe occlusions**. Most of the frame is dominated by large trees and pillars in the foreground, behind which the dog is moving rapidly. This makes the key dynamic regions (dog) subtle and **easily overlooked** in still images visualization in the paper.
>
> Thanks for the reviewer's suggestion, we will highlight these regions using **segmentation masks** in the revised figure and clarify the visual differences in the caption.
>
> **(3) Video Demonstration.** We have also included rendered videos of this case in the supplementary materials, where the differences in motion fidelity and shape consistency among the variants become much more apparent. Please refer to "supp_video/ablation study/libby".
>
> ### **b. Analysis of Variants (Col. 3-5)**
> **(1) Col. 3:** Uses anchors in Global Orientation Field to drive Gaussian deformation. While this captures overall motion direction and recover mainly the **dog's torso**, it lacks local adaptivity and thus fails to reconstruct subtle body parts such as the **legs** and **tail**.
>
> **(2) Col. 4:** Incorporates time as a conditioning variable, allowing temporally-adaptive modeling of motion. This recovers more articulated motion details.
>
> **(3) Col. 5 (full OriGS):** Uses local orientation as an additional condition, enabling the model to adaptively slice the Hyper-Gaussian based on fine-scale changes in motion direction. As a result, **leg and tail movements** involving rapid local directional shifts are reconstructed with much greater fidelity and consistency.
>
>
> # **W2: Presentation of Method Section**
> We sincerely appreciate your feedback regarding the clarity of Section 4.2. We plan to refine this section from two perspectives:
>
> **(1) Add a brief roadmap at the start of Section 4.2 to outline the logical connections of key components.**
> - **Multivariate Modeling of Dynamic States:** We introduce a probabilistic dynamic state to capture local motion variations.
>
> - **Hyper-Gaussian Representation:** We extend 3D Gaussians into higher-dimensional to encode the probabilistic dynamic state.
>
> - **Conditioned Slicing for Dynamic State Inference:** The probabilistic dynamic state is inferred via conditioned slicing, which obtains deformations.
>
> - **Modulation of Hyper-Gaussian Attributes:** The inferred deformations is then used to modulate Gaussian attributes (position, scale, etc.).
>
> - **Anchor-Guided Deformation and Conditioning:** We explain how the conditioning variables in slicing (e.g., orientation) are obtained through interpolation over the Global Orientation Field.
>
> **(2) Restructuring for better flow.** To prevent overloading, we will move the explanation of anchor-based interpolation (currently starting from line 227) earlier into Section 4.1. This provides a self-contained treatment of the Global Orientation Field, while allowing Section 4.2 to focus entirely on the Hyper-Gaussian module.
>
> We appreciate this opportunity to improve clarity. In parallel, we are grateful that multiple reviewers found our exposition largely effective. For example:
> - Reviewer 2Vka noted "the paper is mostly well-written, including rigorous derivation of new Hyper-Gaussians"
> - Reviewer 1JMD commented "enjoyed reading the paper in general; well-written and very clear in its exposition; the method is straightforward and elegant".
>
>
> ## **Q1: Local deformations queried by global anchor positions (as well)**
> We thank the reviewer for this insightful question! While global position may seem intuitively useful, it does not align well with our representation. Our findings are summarized below:
>
> **(1) Slicing in high-dimensional Gaussian favors abstract conditioning.** As formulated in Eq.14, each Hyper-Gaussian is activated most strongly near its learned mean along the conditioning axes. Thus, choosing a conditioning variable that reflects **underlying motion pattern**, rather than **absolute location**, leads to more meaningful and coherent slicing behavior.
>
> **(2) Global anchor position leads to over-localization and reduced generalization.** When using global position as part of the query, the model tends to overfit to spatial anchors. Each primitive becomes tightly "pinned" to its specific location, making it harder to generalize deformation behaviors across regions with similar motion patterns.
>
> **(3) Orientation offers more abstract and transferable motion priors, serving as a generalized indicator of local motion direction.** Conditioning on orientation enables the model to capture **shared** deformation modes across space and time, such as consistent leg motion aligned with walking direction, regardless of **absolute position**. As our task involves modeling dynamic deformation fields, orientation is a natural and expressive cue.

---

> ### Author Response · Authors · 2025-08-04
> **Thank you for your time**
>
> Dear Reviewer QZBh,
>
> Thank you for dedicating your time reviewing our paper. As the discussion period deadline is approaching, we kindly invite any further comments or concerns you might have. Your feedbacks have been immensely valuable to us for refining the paper.
>
> Best,
>
> The Authors

---

> ### Author Response · Authors · 2025-08-06
> **2 DAYS REMAINING: We would like to learn Reviewer's opinion and address any remaining concerns**
>
> Dear Reviewer QZBh,
>
> As there are 2 days remaining for the discussion period, we would kindly like to inquire if you would get a chance to review our response and if there are any remaining questions we can address.
>
> Your insightful feedbacks are crucial for us. We are truly keen to continue a constructive discussion with you to refine our work further.
>
> Best regards,
>
> Authors

---

> ### Author Response · Authors · 2025-08-08
> **1 DAY REMAINING: We would like to learn Reviewer's opinion and address any remaining concerns**
>
> Dear Reviewer QZBh,
>
> As there is 1 day remaining for the discussion period, we would kindly like to inquire if you would get a chance to review our response and if there are any remaining questions we can address.
>
> Your insightful feedbacks are crucial for us. We are truly keen to continue a constructive discussion with you to refine our work further.
>
> Best regards,
>
> Authors

---

### Official Review · Reviewer_2Vka · 2025-07-03

**Clarity:** 3
**Significance:** 3
**Originality:** 4
**Rating:** 4
**Confidence:** 4

**Summary:**

The paper presents a new method for dynamic Gaussians Splatting reconstruction from monocular RGB videos. The main contribution is in making the orientation of motion as first-class citizen of the method. First, a Global Orientation Field is estimated that captures main forward directions of the motion. Then, Orientation-aware Hyper-Gaussians are defined that can model complex local motion in a robust way. Quantitative results show improvement over previous methods.

**Questions:**

Overall, I view the paper and the idea novel and original. However, the limited experimental validation prevents me from giving a higher score. Additional evaluation on 2D/3D tracking accuracy could be valuable. Moreover, the evaluation on DyCheck is questionable, and it's not clear if the comparison is fair. The qualitative results are not satisfactory.

**Ethical Concerns:**

["NO or VERY MINOR ethics concerns only"]

**Final Justification:**

I appreciate additional experiments on 2D point tracking. It would still be beneficial for the paper to include evaluation on 3D point tracking and dynamic mask segmentation tasks. I still view the paper as a borderline one, but I'm willing to raise my score to Borderline Accept.

**Limitations:**

Yes, the limitations are discussed in the supplementary files.

**Paper Formatting Concerns:**

No concerns.

**Quality:**

2

**Strengths And Weaknesses:**

Strengths:
* An idea of basing everything on the motion orientation is indeed novel and interesting.
* A new type of Gaussian Splats are proposed, called Orientation-aware Hyper-Gaussian.
* The paper is mostly well-written, including rigorous derivation of new Hyper-Gaussians.
* The proposed method improves the novel-view synthesis results on the DyCheck dataset.
* Ablation study supports some design choices.

Weaknesses:
* The supplementary video is hard to explore as the folder contains many subfolders and many different videos. It makes it almost impossible to compare different methods. Instead, the authors should have prepared a single video with side-by-side comparisons. Regarding the reconstruction quality in the video: it seems quite low-quality, and the qualitative results are underwhelming.
* Only novel view synthesis evaluation is presented. Why wasn't the 2D/3D tracking accuracy evaluated (as in Shape-of-Motion)? Dynamic mask segmentation could have also been evaluated.
* In Table 1, were all the methods run by the authors of this paper, or were the numbers just copied from the papers? This is very important, because some papers use different subsets of the DyCheck dataset (e.g. Shape-of-Motion doesn't use all the scenes in Table 1), so if some numbers were just copied, the comparison is invalid. And given that the results in Table 1 for SoM do not include per-scene scores, most likely the number were copied.

---

> ### Author Rebuttal · Authors · 2025-07-30
>
> # **W1: Supplementary Video**
> We sincerely thank the reviewer for the suggestion on organizing the supplementary materials! We realize that the current folder-based structure may hinder efficient comparison. We will restructure the supplementary materials to showcase side-by-side comparisons in the revised version.
>
> Regarding the perceived reconstruction quality, we would like to provide further context to better frame our results:
>
>
> **(1) Challenge of Monocular 4D Reconstruction.** 4D reconstruction from casual monocular videos is highly ill-posed and under-constrained. This challenge is particularly acute when rendering novel views under spatiotemporal configurations that differ **significantly** from training views, such as rotating the camera at frozen time or introducing aggressive viewpoint movements, as demonstrated in our videos. These scenarios require models to generalize beyond observed motion paths and to complete occluded or unobserved geometry, often with minimal depth cues. While current methods [1,2,3,4] may produce visually appealing results under **mild** view changes, they typically struggle with such out-of-distribution (OOD) conditions, leading to collapse in visual quality.
>
> We believe it is important to evaluate reconstruction methods not only in easy settings but also under these more extreme cases to distinguish the advancement. Therefore, rather than claiming perfection in absolute quality, our goal is to push the boundary of what is achievable under casual, monocular, and unconstrained capture.
>
> **(2) Comprehensive Evaluation and Method Robustness.** Our supplementary video provides comprehensive qualitative results, including standard training views, rotating camera at frozen time, and moving camera across video. These three protocols range from in-distribution to highly challenging OOD conditions. In all cases, our method demonstrates clear advantages over existing SOTA approaches. This robustness is further suggested by qualitative superiority in side-by-side comparisons (Figure 3) and quantitative gains on DyCheck (Table 1).
>
> **(3) Peer Validation.** We would also like to highlight that multiple reviewers acknowledged our method's superiority. For instance, Reviewers p9KW highlighted "results are outperforming all other methods on DyCheck", Reviewer QZBh observed "the proposed method surpasses existing baselines", and Reviewer 1JMD appreciated that "the method is straightforward and elegant with good results". We believe these remarks reflect a broader recognition of our contributions.
>
>
> # **W2: More Evaluation**
> We thank the reviewer for the helpful suggestion.
> ### **a. Point Tracking**
> We have extended our experiments to include dense 2D point tracking on the DyCheck benchmark, using the provided PCK-T@5% metric (percentage of correctly tracked keypoints at a 5% interval), which is adopted in recent works [1,4]. The results are shown below:
>
> | Methods             | PCK-T ↑ |
> |---------------------|---------|
> | Nerfies             | 0.400   |
> | HyperNeRF           | 0.453   |
> | Dyn. Gauss.         | 0.079   |
> | 4D Gauss.           | 0.073   |
> | CoTracker           | 0.803   |
> | Gauss. Marbles      | 0.806   |
> | BootsTAPIR          | 0.779   |
> | MoSca               | 0.824   |
> | **Ours**            | **0.851**  |
>
> Our OriGS significantly outperforms prior approaches in point tracking, further validating the strength of our orientation-anchored representation in capturing complex motion patterns.
>
> ### **b. Dynamic Mask Segmentation**
> To the best of our knowledge, this is not a standard evaluation protocol in the literature for dynamic scene reconstruction [1,2,3,4]. Most recent methods focus on view synthesis but do not incorporate semantic mask evaluation. Nevertheless, we agree that evaluating segmentation is an interesting direction, and our OriGS could potentially benefit such tasks.
>
>
> # **W3: Validity of Table 1 Comparisons**
> We thank the reviewer for pointing out the potential ambiguity regarding Table 1. We would like to clarify that the comparisons presented are **fair and valid**.
>
> **(1)** All methods in Table 1 were evaluated over the same scene set, following recent work like MoSca [4]. For consistency, we adopted the same evaluation protocol as in MoSca and **included the average result of Shape-of-Motion (SoM) as reported in the MoSca paper.**
>
> **(2)** While we do not report per-scene results for SoM (since they do not release preprocessed data for the DyCheck scenes we use), the reported average result is consistent and directly comparable, given that it **comes from the exact same subset** used across all methods.
>
> We will clarify this point in the main paper to ensure transparency. We hope this explanation resolves the reviewer's concern about the validity of the comparison.
>
> [1] Dynamic Gaussian Marbles for Novel View Synthesis of Casual Monocular Videos.
>
> [2] Shape of Motion: 4D Reconstruction from a Single Video.
>
> [3] MoDGS: Dynamic Gaussian Splatting from Casually-captured Monocular Videos with Depth Priors.
>
> [4] MoSca: Dynamic Gaussian Fusion from Casual Videos via 4D Motion Scaffolds.

---

> > ### Comment · Reviewer_2Vka · 2025-08-04
> >
> > Thank you for your answers. I appreciate additional experiments on 2D point tracking. It would still be beneficial for the paper to include evaluation on 3D point tracking and dynamic mask segmentation tasks. I still view the paper as a borderline one, but I'm willing to raise my score to Borderline Accept.

---

> > > ### Author Response · Authors · 2025-08-04
> > > **Thank you**
> > >
> > > Dear Reviewer 2Vka,
> > >
> > > Thank you very much for your thoughtful feedback and engagement during the review process.
> > >
> > > Best regards,
> > >
> > > Authors

---

> ### Author Response · Authors · 2025-08-04
> **Thank you for your time**
>
> Dear Reviewer 2Vka,
>
> Thank you for dedicating your time reviewing our paper. As the discussion period deadline is approaching, we kindly invite any further comments or concerns you might have. Your feedbacks have been immensely valuable to us for refining the paper.
>
> Best,
>
> The Authors

---

### Official Review · Reviewer_p9KW · 2025-07-03

**Clarity:** 2
**Significance:** 4
**Originality:** 3
**Rating:** 5
**Confidence:** 5

**Summary:**

This is a paper on 4D reconstruction. The basic concept of the paper is that orientation of scene components changes smoothly due to physics constraints like preservation of the angular momentum.
To achieve this the authors introduce the notions of the Global Orientation Field and the Orientation-aware Hyper-Gaussian.
An initial dominant orientation is derived from the PCA using the temporal mean of trajectories over an initial window. Propagation happens through Procrustes on the mean compensated trajectories. A state consisting of position and orientation deviations, time, and the field orientation is modeled as a Gaussian distribution. The state components are dependent, the deviations are conditioned in time and field orientation. Deformation is modeled through anchors as in the Embedded Deformation Graph frameworks.
Trajectory estimation leverages foundation models on depth and tracking. Bundle adjustment is applied to compute camera intrinsics and extrinsics.
Experiments include evaluation on DAVIS, SORA, VOS, and DyCheck.

**Questions:**

Q1: Why does Procruses guarantee a smoothness? Is it because of the overlap in temporal windows? One would expect that a Kalman Filter would be more effective on that.

Q2: Explain the difference between the anchors in this paper and the scaffolds used in MOSCA [36].

Q3: It is awkard to include time in a state vector (reminds more of Galilean or Lorentz transformations), please explain why.

Q4: It is not clear what the authors mean with hyper-state in line 192, isn't it just the mean of the states?

Q4: Explain the special minus sign in line 209 and eq. (14), what is its exact definition? (the special plus in line 221 is clear because it refers to quaternions).

Q5: Loss function belongs to the main text not the appendix.

Q6: Which exactly foundation models are used for depth and 2D correspondence (many are listed in l. 35 of the appendix).

Q6: Please explain the ablation. The jump from Deform with GOF to Hyper-Gaussian with t does include only the incorporation of t but no probabilistic formulation?

Q7: The performance drops only slightly when bundle adjustment is used to estimate poses rather than the ground truth (behavior observed in MOSCA) as well. This is surprising given that camera motions are quite ambiguous and one would expect that an out of the box method like BARF would introduce errors.

Elaborate on the weaknesses.

**Ethical Concerns:**

["NO or VERY MINOR ethics concerns only"]

**Final Justification:**

The authors made a significant effort in clarifying the mathematics of the approach and I hope that they will incorporate it in their camera ready. The writing would have been clearer if the authors had attended a class on differential geometry which I really recommend them to do as soon as possible.

**Limitations:**

yes

**Quality:**

3

**Strengths And Weaknesses:**

S1: The main strength of the paper is in the introduction of a dynamics model based on the smoothness of orientation.

S2: Results are outperforming all other methods on DyCheck.

S3: The ablation is helpful but has to be better explained (see Q below).

I see only few major weaknesses in the paper (the rest will be in questions) and I am willing to increase the score.

W1: The introduction of time in the state is intriguing but there is no prior in the science literature of not relativistic systems. There is no 3D object or other dynamic tracking that includes time and there is no motivation given. Does this result in a time-variant system? How can we have the time in the state when it appears also as a superscipt in the propagation?

W2: The conditioning of the deviation on the dominant orientation (9) is also not explained.
At this point the reader wonders if $O^{t'}$ is one per scene or a real field.
I think that (9-11) could be easily written in a clean Kalman Filter formulation.

W3: Because of the introduction of probabilities the math became very messy. Several covariances appear that are not defined. Is the covariance with respect to $\Delta O$ based on quaternions?
How is a Gaussian defined on the SE(3) manifold? What does a covariance in time mean (is there noise in time?). Pages 5 and 6 need significant mathematical re-work.

Very minor weakness: $VU^T$ is in $O(3)$ it has to be multiplied with diag(1,1,det(UV^T)) to make it SO(3).

---

> ### Author Rebuttal · Authors · 2025-07-30
>
> # **W1 & Q3: Time in the State**
>
> ### **a. Motivation**
> **(1) Practical Goal.** Our motivation for introducing time (alongside orientation) is not to formulate a relativistic system, but to capture temporally varying motion patterns in a practical way for 4D reconstruction. By jointly conditioning on orientation and time, our model enables time-adaptive deformation in a unified slicing-based framework.
>
> **(2) Disambiguating Orientations.** While orientation provides valuable cues, it may not fully resolve temporal ambiguity. In real-world scenes, identical or similar orientations can arise at different time, yet the local dynamics or scene context can differ. For instance, an object returning to a familiar orientation may still be undergoing a different trajectory or surrounding geometry, depending on the phase of motion. Incorporating time allows the model to distinguish such cases and adapt to the appropriate context.
>
>
> ### **b. Time in the State & Superscipt in Propagation**
> Time $t'$ is involved in two processes:
>
> **(1) Indexing Global Orientation Field.** We propagate the initial orientations through the video to build a Global Orientation Field. The superscript of $O^{t'}$ denotes that the orientation field is queried at frame $t'$.
>
> **(2) Conditioning Hyper‑Gaussian.** Time $t'$ is included in the conditioning variable for deformation.
>
> The workflow is sequential and consistent: we first query the orientation field to obtain $O^{t'}$, and then use both $t'$ and $O^{t'}$ in conditioned slicing for local deformation.
>
>
> ### **c. Validation from Prior Studies**
> Recent advances in dynamic scene modeling, such as 4DGS [1] and 4D-Rotor GS [2], have shown the benefits of embedding time into Gaussian primitives for learning spatio-temporal patterns (line 85 in our paper).
>
> Our method is conceptually aligned with this direction, but extends it through a Hyper-Gaussian, modeling space, time, geometry, and orientation in a unified representation.
>
>
> # **W2: Clarifying $O^{t'}$**
> **(1) $O^{t'}$ is a Field.** In Eq.9, $O^{t'}$ refers to the local orientation queried from Global Orientation Field at a specific 3D position and time step $t'$ (line 228 & 240). For each Hyper‑Gaussian, we first deform its position and rotation by **Eq.15**; we then query the orientation field at the deformed position and $t'$ to obtain a unique $O_{t'}$ by **Eq.16**. We will explain the query mechanism earlier (around line 205).
>
>
> **(2) Kalman Filter.** We appreciate the insightful suggestion and agree that Kalman Filter offers a principled perspective to our framework. We believe Kalman-based approach could provide additional benefits, such as refining motion consistency or modeling uncertainty in dynamic scenes. We consider this a promising avenue for future exploration.
>
>
> # **W3: Clarification on Math**
> ### **a. Covariances Not Defined**
> We define a multivariate Gaussian: $\xi = (\Delta p, \Delta g, t, O).$ The covariance matrix $\Sigma_{\xi}$ captures the correlation among these variables. All covariance blocks appearing in Eq.8–14, such as $\Sigma_{(t,O)}$ and $\Sigma_{(\Delta p,\Delta g),(t,O)}$, are **submatrices** of $\Sigma_{\xi}$, which select the rows and columns corresponding to the respective variable groups in the subscript.
>
>
> ### **b. Covariance in $O$**
> The orientation $O \in \mathrm{SO}(3)$ is modeled using quaternions. When performing conditioned slicing (e.g., Eq.10), we adopt the logarithmic map and define the relative rotation $O^{t'} \ominus \mu_O$ as a tangent vector in $\mathfrak{so}(3)$ (Please also refer to **Q4b: Special Minus Sign** below). Thus, the covariance over $O$ lies in the tangent space at $\mu_O$, and we operate in Euclidean coordinates during slicing.
>
>
> ### **c. Gaussian on Manifold**
> Our formulation does not place a Gaussian directly on the full $SE(3)$ manifold. Instead, the dynamic state is modeled on a product space. The Euclidean components are handled with a standard multivariate Gaussian, while the rotational component is handled via a quaternion-based relative rotation followed by a logarithmic map into the tangent space $\mathfrak{so}(3)$. This Euclidean approximation allows us to apply closed-form Gaussian conditioning.
>
> **Established Principle.** Recent high-dimensional Gaussian splatting methods, such as N-DG [3] and 6DGS [4], embed additional cues into Euclidean spaces and perform Gaussian conditioning accordingly. Specifically, N-DG augments Gaussians with albedo and roughness, while 6DGS integrates view direction to model view-dependent effects. Building on this principled strategy, we extend the formulation to model space, time, geometry, and orientation in a unified representation.
>
>
> ### **d. Covariance in Time**
> Including time during slicing allows each Hyper‑Gaussian to modulate its response by temporal proximity, acting as a soft activation kernel along the time axis. While we do not explicitly model "noise in time", including time as a slicing dimension naturally gives rise to a conditional variance along the temporal axis. Intuitively, its covariance serves to regulate how temporally sensitive the dynamic state is to deviations in the target time.
>
>
> # **W4: $VU^T$**
> In our implementation, we have applied the standard correction step to make it $SO(3)$. We will clarify this.
>
>
> ---
>
>
> # **Q1: Procruses**
> **(1) Clarification.** Our use of Procrustes is not intended to explicitly optimize for smoothness. Instead, we estimate the initial orientation over a temporal window in early frames, and propagate it via frame-to-frame Procrustes from a geometric perspective.
>
> **(2) Implicit Smoothness.** While not designed to enforce smoothness, we perform Procrustes within local neighborhoods (line 153), avoiding abrupt discontinuities. Additionally, we apply interpolation (line 241) over surrounding anchors to support arbitrary querying, which introduces smoothness.
>
> **(3) Kalman Filter.** We acknowledge that Extended Kalman Filter on $SO(3)$ offers a principled alternative, particularly under noisy or ambiguous motion. We view this as a promising direction for future exploration.
>
>
> # **Q2: Difference with Scaffold**
> Our oriented anchors and MoSca's scaffolds share conceptual similarities, as both are derived from Embedded Deformation Graphs. However, there are important differences:
>
> **(1) Lack of Orientation in Scaffold.** Scaffold nodes are not designed to encode the actual motion orientation. Their "orientation" are initialized as identity matrices and only **relative** rotations are estimated between adjacent nodes over time.
>
> **(2) Scaffold** serves as a deformation support structure to drive 3D Gaussian primitives, while relying heavily on the primitives themselves to capture complex local dynamics. **Oriented anchors** capture the forward direction across time, serving as conditioning signal for local deformation.
>
>
> # **Q3: Time in the State**
> Please see **W1 & Q3: Time in the State** above.
>
>
> # **Q4a: Hyper-state**
> Sorry for the confusion. The "hyper-state" in lines 192 and 222 should simply be "state", as it is just the mean of the states.
>
>
> # **Q4b: Special Minus Sign**
> The $\ominus$ aims to compute the relative rotation between two orientations. Given two unit quaternions $O^{t'}$ and $\mu_O$, the relative rotation is:
> $$q_{\text{rel}} = O^{t'} \cdot \mu_O^{-1}.$$
> Because a unit quaternion's inverse is simply its conjugate, this can be calculated efficiently by:
> $$q_{\text{rel}} := O^{t'} \cdot \overline{\mu_O}.$$
> To incorporate this relative rotation into our conditioned slicing, we next apply the standard $SO(3)$ logarithmic map. The $\ominus$ is then defined as:
> $$O^{t'} \ominus \mu_O = \log(q_{\text{rel}}) \in \mathfrak{so}(3).$$
> This produces an axis-angle vector representing the minimal rotation from $\mu_O$ to $O^{t'}$, and ensures that the result lies in an Euclidean space suitable for Gaussian slicing.
>
>
> # **Q5: Loss Function**
> We will move the optimization detail to main paper as Section 4.3.
>
>
> # **Q6a: Foundation Models**
> We use DepthCrafter [5] and SpatialTracker [6].
>
>
> # **Q6b: Clarifying Ablation**
> The variant "Hyper-Gaussian with t" still involves a probabilistic formulation. Specifically:
>
> - This variant defines a Hyper-Gaussian over $\xi = (\Delta \mathbf{p}, \Delta \mathbf{g}, t)$ (line 304).
>
> - At inference, we perform conditioned slicing on the Hyper-Gaussian to obtain time-dependent deformation, following the mechanism in Eq.9–14.
>
> - This variant removes orientation from both the state modeling and the conditioning process, thus relying solely on time for deformation inference.
>
>
> # **Q7: Slight Performance Drops**
> Thank you for pointing out this interesting observation. We agree that one might intuitively expect larger degradation with estimated poses, especially in monocular settings where camera motion can be ambiguous.
>
> However, we find that this phenomenon is closely tied to the **design of DyCheck benchmark** [7]. DyCheck is specifically constructed to evaluate 4D reconstruction under casual monocular conditions, featuring:
> - **Slow camera motion** to preserve the monocular setting
> - **Fast and complex object motion** for scene-level dynamic challenges
>
> The major bottleneck on DyCheck is not camera ambiguity, but rather the difficulty of modeling complex scene dynamics. Thus, even when using poses estimated, the performance drop may be relatively slight, as the limiting factor lies elsewhere.
>
>
> ---
> [1] Real-time Photorealistic Dynamic Scene Representation and Rendering with 4D Gaussian Splatting.
>
> [2] 4D-Rotor Gaussian Splatting: Towards Efficient Novel View Synthesis for Dynamic Scenes.
>
> [3] N-Dimensional Gaussians for Fitting of High Dimensional Functions.
>
> [4] 6DGS: Enhanced Direction-Aware Gaussian Splatting for Volumetric Rendering.
>
> [5] DepthCrafter: Generating Consistent Long Depth Sequences for Open-World Videos.
>
> [6] SpatialTracker: Tracking Any 2D Pixel in 3D Space.
>
> [7] Monocular Dynamic View Synthesis: A Reality Check.

---

> > ### Author Response · Authors · 2025-08-04
> > **Thank you for your time**
> >
> > Dear Reviewer p9KW,
> >
> > Thank you for dedicating your time reviewing our paper. As the discussion period deadline is approaching, we kindly invite any further comments or concerns you might have. Your feedbacks have been immensely valuable to us for refining the paper.
> >
> > Best,
> >
> > The Authors

---

> > ### Comment · Reviewer_p9KW · 2025-08-05
> > **Thank you for answering all my questions**
> >
> > W1/Q3: I accept the argument of the authors that the introduction of time is not meant in a dynamical systems sense but more as a means to become more discriminative rather than invariant in time.
> >
> > W3: The authors’ response is not rigorous. They introduced a probabilistic aspect in (9)-(11) and they have to derive the equations.
> >
> > Q4b: There is an established way to write quaternion operations without introduction of new operators.
> >
> > I really appreciate all other responses with which I am very satisfied.
> > I think the authors could have been (or will do in the camera ready) more meticulous in the math and in the description of the methods (losses, foundation models etc).

---

> > > ### Author Response · Authors · 2025-08-05
> > > **Thank you for your feedback**
> > >
> > > Thank you for the constructive follow-up comments and for acknowledging the other parts of our rebuttal. To clarify the remaining concerns raised in **W3** and **Q4b**, we would like to detail the derivation of Eq.9–11.
> > >
> > > # **W3: Derivation of Eq.9-11**
> > > ### **a. State Modeling**
> > > While the dynamic state is originally defined on a product manifold: $\xi = (\Delta \mathbf{p}, \Delta \mathbf{g}, t, O),$ we perform probabilistic inference in a Euclidean space by leveraging the tangent space of $\mathrm{SO}(3)$.
> > >
> > >
> > > ### **b. Logarithmic Mapping for Orientation**
> > > To enable conditioning with respect to orientation, we apply the logarithmic map to express the relative rotation from the mean $\mu_O$ to the query orientation $O^{t'}$:
> > >
> > > $$
> > > \delta_O := O^{t'} \ominus \mu_O = \log\left(O^{t'} \cdot \overline{\mu_O}\right) \in \mathfrak{so}(3).
> > > $$
> > >
> > > This yields an axis-angle vector in Euclidean space, suitable for Gaussian slicing.
> > >
> > >
> > > ### **c. Formulation of Gaussian Slicing**
> > > Let the slicing vector be:
> > >
> > > $$
> > > x_{\text{euc}} := \begin{bmatrix}\delta_t \\\ \delta_O\end{bmatrix} = \begin{bmatrix}t' - \mu_t \\\ log\left(O^{t'} \cdot \overline{\mu_O}\right)\end{bmatrix}.
> > > $$
> > >
> > > We consider the covariance over $y = \left(\Delta \mathbf{p}, \Delta \mathbf{g}\right)$ and $x = x_{\text{euc}}$:
> > >
> > > $$
> > > \begin{bmatrix}
> > > y \\\\ x
> > > \end{bmatrix}
> > > \sim \mathcal{N}
> > > \left(
> > > \begin{bmatrix}
> > > \mu_y \\\\
> > > \mu_x
> > > \end{bmatrix},
> > > \begin{bmatrix}
> > > \Sigma_{yy} & \Sigma_{yx} \\\\
> > > \Sigma_{xy} & \Sigma_{xx}
> > > \end{bmatrix}
> > > \right),
> > > $$
> > >
> > > These blocks are the "$\Sigma_{(\Delta p,\Delta g),(t,O)}$" etc. referenced in Eq.8–11; each is the covariance corresponding to the variable groups in the subscript. After applying the log map, the orientation component of the covariance refers to the (Euclidean) tangent space at $\mu_O$, where probabilistic operations are valid.
> > >
> > > Built upon this, we apply the standard conditioning formula:
> > >
> > > $$
> > > P(y \mid x) = \mathcal{N}\left(
> > > \mu_y + \Sigma_{yx} \Sigma_{xx}^{-1} x_{\text{euc}},\;
> > > \Sigma_{yy} - \Sigma_{yx} \Sigma_{xx}^{-1} \Sigma_{xy}
> > > \right).
> > > $$
> > >
> > >
> > > # **Q4b: Notation for Quaternion Operations**
> > > To avoid introducing an ad-hoc "$\ominus$", we will adopt the conventional expression of logarithmic map:
> > >
> > > $$
> > > \delta_O = \log(O^{t'}\,\mu_O^{-1}).
> > > $$
> > >
> > > In our revised explanation, we will remove "$\ominus$" entirely and replaced it with established quaternion operations.
> > >
> > >
> > > # **Additional Revisions in the Paper**
> > > - We will add an "Implementation Details" paragraph specifying foundation models, loss terms, and optimization strategy in the main paper.
> > > - Footnotes will be inserted near Eq.8 and Eq.10 to remind the reader that "orientation covariances are evaluated in the tangent space".
> > > - We will detail the full derivation of Eq.9-11 in supplementary materials.
> > > - Quaternion operations in the equations will follow the established form.
> > >
> > > Thank you again for your insightful feedback.

---

> > > ### Author Response · Authors · 2025-08-08
> > > **1 DAY REMAINING: We would like to learn Reviewer's opinion and address any remaining concerns**
> > >
> > > Dear Reviewer p9KW,
> > >
> > > As there is 1 day remaining for the discussion period, we would kindly like to inquire if you would get a chance to review our response and if there are any remaining questions we can address.
> > >
> > > Your insightful feedbacks are crucial for us. We are truly keen to continue a constructive discussion with you to refine our work further.
> > >
> > > Best regards,
> > >
> > > Authors

---

### Author Response · Authors · 2025-08-02
**Follow-up on Discussion**

Dear Reviewers and Area Chair,

We hope this message finds you well.

We have carefully addressed all reviewer comments in our responses. As the discussion period draws to a close on August 6, we would be grateful for any further questions or concerns you may wish to raise. We are fully committed to engaging with any additional feedback during this final stage.

Thank you very much for your time and thoughtful consideration.

Best regards,

The Authors

---

### Comment · Area_Chair_3QvJ · 2025-08-04
**Quick Reminder: NeurIPS 2025 Author Reviewer Discussion**

Dear Reviewers,

This is a friendly reminder to please engage in the author discussion phase if you haven’t already. Your responses to author replies are important and appreciated. The discussion period ends on Aug 6, so we encourage you to participate soon. Thank you for your contributions to NeurIPS 2025~

Best regards,

AC

---

### Note · Authors · 2025-08-15

Dear ACs, SACs, and reviewers,

We sincerely thank all reviewers and the area chair for their time and efforts. We are pleased to see that **Reviewers p9KW and 1JMD expressed satisfaction** with our rebuttal and had no further questions, and that **Reviewer 2Vka raised the score** after our clarifications. **Reviewer QZBh hasn't respond to us during the discussion** and we hope our rebuttal has address the questions.

Our rebuttal and planned revisions have been strengthened by the reviewers’ suggestions:

- Reviewer p9KW – Expanded discussion on the motivation and derivation of our state modeling equations.
- Reviewer 2Vka – Additional evaluation on point tracking and detailed description of our evaluation protocol.
- Reviewer QZBh – Clarification of ablation study results and method presentation, and discussion on global anchor position querying.
- Reviewer 1JMD – Additional visualization, and discussion on sampling-based method.

**Based on our rebuttal, we have largely addressed concerns from Reviewers p9KW, 2Vka, and 1JMD, and received their recognition.** **Unfortunately, Reviewer QZBh has not participated further in the discussion.**

**We also noticed that Reviewer QZBh rated the paper with the lowest score on clarity (1: poor), which stands in contrast to other reviewers’ assessments.** For example,

- Reviewer 2Vka noted "the paper is mostly **well-written**, including **rigorous derivation** of new Hyper-Gaussians"
- Reviewer 1JMD commented "**enjoyed reading** the paper in general; **well-written** and **very clear** in its exposition; the method is **straightforward** and **elegant**".

This suggests that QZBh’s difficulty in following our qualitative ablation results and methodology may reflect differences in interpretation, rather than an intrinsic issue in the paper, especially given the consensus from the other reviewers.

Best regards,

Authors

---

### Decision · Program_Chairs · 2025-09-17

**Decision:**

Accept (poster)

**Comment:**

This paper presents OriGS, a novel framework for high-quality 4D reconstruction that introduces an elegant orientation-anchored representation to model complex scene dynamics from monocular videos. The work received strong support from the reviewers, who were convinced by its originality, superior performance on challenging benchmarks, and the clarity of its formulation. The authors' thorough rebuttal, including new experiments, successfully addressed initial concerns and solidified the consensus for acceptance.